# Amortized Proximal Optimization

**Juhan Bae**[*1,2], **Paul Vicol**[*1,2], **Jeff Z. HaoChen**[3], **Roger Grosse**[1,2]
[1]University of Toronto, [2]Vector Institute, [3]Stanford University
{jbae, pvicol, rgrosse}@cs.toronto.edu
jhaochen@stanford.edu

## Abstract

We propose a framework for online meta-optimization of parameters that govern optimization, called Amortized Proximal Optimization (APO). We first interpret various existing neural network optimizers as approximate stochastic proximal point methods which trade off the current-batch loss with proximity terms in both function space and weight space. The idea behind APO is to amortize the minimization of the proximal point objective by meta-learning the parameters of an update rule. We show how APO can be used to adapt a learning rate or a structured preconditioning matrix. Under appropriate assumptions, APO can recover existing optimizers such as natural gradient descent and KFAC. It enjoys low computational overhead and avoids expensive and numerically sensitive operations required by some second-order optimizers, such as matrix inverses. We empirically test APO for online adaptation of learning rates and structured preconditioning matrices for regression, image reconstruction, image classification, and natural language translation tasks. Empirically, the learning rate schedules found by APO generally outperform optimal fixed learning rates and are competitive with manually tuned decay schedules. Using APO to adapt a structured preconditioning matrix generally results in optimization performance competitive with second-order methods. Moreover, the absence of matrix inversion provides numerical stability, making it effective for low-precision training.

## 1 Introduction

Many optimization algorithms widely used in machine learning can be seen as approximations to an idealized algorithm called the proximal point method (PPM). When training neural networks, the stochastic PPM iteratively minimizes a loss function $\mathcal{J}_\mathcal{B} \colon \mathbb{R}^m \to \mathbb{R}$ on a mini-batch $\mathcal{B}$, plus a proximity term that penalizes the discrepancy from the current iterate:

$$\boldsymbol{\theta}^{(t+1)} \leftarrow \operatorname*{arg\,min}_{\mathbf{u} \in \mathbb{R}^m} \mathcal{J}_{\mathcal{B}^{(t)}}(\mathbf{u}) + \lambda D(\mathbf{u}, \boldsymbol{\theta}^{(t)}), \tag{1}$$

where $D(\cdot, \cdot)$ measures the discrepancy between two vectors and $\lambda > 0$ is a hyperparameter that controls the strength of the proximity term. The proximity term discourages the update from excessively changing the parameters, hence preventing aggressive updates. Moreover, the stochastic PPM has good convergence properties [4]. While minimizing Eq. 1 exactly is usually impractical (or at least uneconomical), solving it approximately (by taking first or second-order Taylor series approximations to the loss or the proximity term) has motivated important and widely used optimization algorithms such as natural gradient descent [1] and mirror descent [7]. Stochastic gradient descent (SGD) [69] itself can be seen as an approximate PPM where the loss term is linearized and the discrepancy function is squared Euclidean distance.

---

[*]Equal Contribution

36th Conference on Neural Information Processing Systems (NeurIPS 2022).

Inspired by the idea that the PPM is a useful algorithm to approximate, we propose to amortize the minimization of Eq. 1 by defining a parametric form for an update rule which is likely to be good at minimizing it and adapting its parameters with gradient-based optimization. We consider adapting optimization hyperparameters (such as the learning rate) for existing optimizers such as SGD and RMSprop [78], as well as learning structured preconditioning matrices. By choosing a structure for the update rule inspired by existing optimizers, we can take advantage of the insights that went into those optimizers while still being robust to cases where their assumptions (such as the use of linear or quadratic approximations) break down. By doing meta-descent on the optimization parameters, we can amortize the cost of minimizing the PPM objective, which would otherwise take many steps per parameter update. Hence, we call our approach *Amortized Proximal Optimization (APO)*.

Eq. 1 leaves a lot of freedom for the proximity term. We argue that many of the most effective neural network optimizers can be seen as trading off two different proximity terms: a *function space discrepancy (FSD)* term which penalizes the average change to the network's predictions, and a *weight space discrepancy (WSD)* term which prevents the weights from moving too far, encouraging smoothness to the update and maintaining the accuracy of second-order approximations. Our meta-objective includes both terms.

Our formulation of APO is general, and can be applied to various settings, from optimizing a single optimization hyperparameter to learning a flexible update rule. We consider two use cases that cover both ends of this spectrum. At one end, we consider the problem of adapting learning rates of existing optimizers, specifically SGD, RMSprop, and Adam [19]. The learning rate is considered one of the most essential hyperparameters to tune [10], and good learning rate schedules are often found by years of trial and error. Empirically, the learning rate schedules found by APO outperformed the best fixed learning rates and were competitive with manual step decay schedules.

Our second use case is more ambitious. We use APO to learn a preconditioning matrix, giving the update rule the flexibility to represent second-order optimization updates such as Newton's method, Gauss-Newton, or natural gradient descent. We show that, under certain conditions, the optimum of our APO meta-objective with respect to a full preconditioning matrix coincides with damped versions of natural gradient descent or Gauss-Newton. While computing and storing a full preconditioning matrix for a large neural network is impractical, various practical approximations have been developed. We use APO to meta-learn a structured preconditioning matrix based on the EKFAC optimizer [24]. APO is more straightforward to implement in current-day deep learning frameworks than EKFAC and is also more computationally efficient per iteration because it avoids the need to compute eigen-decompositions. Empirically, we evaluate APO for learning structured preconditioners on regression, image reconstruction, image classification, and neural machine translation tasks. The preconditioning matrix adapted by APO achieved competitive convergence to other second-order optimizers.

## 2 Preliminaries

Consider a prediction problem from some input space $\mathcal{X}$ to an output space $\mathcal{T}$. We are given a finite training set $\mathcal{D}_{\text{train}} = \{(\mathbf{x}^{(i)}, \mathbf{t}^{(i)})\}_{i=1}^{N}$. For a data point $(\mathbf{x}, \mathbf{t})$ and parameters $\boldsymbol{\theta} \in \mathbb{R}^m$, let $\mathbf{y} = f(\mathbf{x}, \boldsymbol{\theta})$ be the prediction of a network parameterized by $\boldsymbol{\theta}$ and $\mathcal{L}(\mathbf{y}, \mathbf{t})$ be the loss. Our goal is to minimize the cost function:

$$\mathcal{J}(\boldsymbol{\theta}) = \frac{1}{N} \sum_{i=1}^{N} \mathcal{L}(f(\mathbf{x}^{(i)}, \boldsymbol{\theta}), \mathbf{t}^{(i)}). \tag{2}$$

We use $\mathcal{J}_{\mathcal{B}}(\boldsymbol{\theta})$ to denote the mean loss on a mini-batch of examples $\mathcal{B} = \{(\mathbf{x}^{(i)}, \mathbf{t}^{(i)})\}_{i=1}^{B}$. We summarize our notation in Appendix A.

## 3 Proximal Optimization and Second-Order Methods: A Unifying Framework

We first motivate the proximal objective that we use as the meta-objective for APO, and relate it to existing neural network optimization methods. Our framework is largely based on Grosse [25], to which readers are referred for a more detailed discussion.

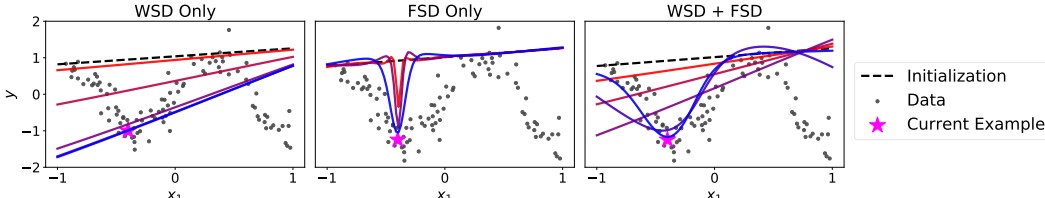

Figure 1: 1D illustration of the exact proximal update on a regression problem with a batch size of 1, inspired by Grosse [25]. The weight of the discrepancy term(s) ($\lambda_{\text{WSD}}$ and $\lambda_{\text{FSD}}$) is decreased from **red** to **blue**.

## 3.1 Proximal Optimization

When we update the parameters on a mini-batch of data, we would like to reduce the loss on that mini-batch, while not changing the predictions on previously visited examples or moving too far in weight space. This motivates the following proximal point update:

$$\boldsymbol{\theta}^{(t+1)} \leftarrow \underset{\mathbf{u} \in \mathbb{R}^m}{\arg\min} \, \mathcal{J}_{\mathcal{B}^{(t)}}(\mathbf{u}) + \lambda_{\text{FSD}} \mathbb{E}_{\tilde{\mathbf{x}} \sim \mathcal{D}}[D_{\text{F}}(\mathbf{u}, \boldsymbol{\theta}^{(t)}, \tilde{\mathbf{x}})] + \lambda_{\text{WSD}} D_{\text{W}}(\mathbf{u}, \boldsymbol{\theta}^{(t)}), \tag{3}$$

where $D_{\text{F}}(\mathbf{u}, \boldsymbol{\theta}, \mathbf{x}) = \rho(f(\mathbf{x}, \mathbf{u}), f(\mathbf{x}, \boldsymbol{\theta}))$ and $D_{\text{W}}(\mathbf{u}, \boldsymbol{\theta}) = 1/2\|\mathbf{u} - \boldsymbol{\theta}\|_2^2$ are discrepancy functions (described below). Here, $\lambda_{\text{FSD}}$ and $\lambda_{\text{WSD}}$ are hyperparameters that control the strength of each discrepancy term, $\tilde{\mathbf{x}}$ is a random data point sampled from the data-generating distribution $\mathcal{D}$, and $\rho(\cdot, \cdot)$ is the output-space divergence.

The proximal objective in Eq. 3 consists of three terms. The first term is the loss on the current mini-batch. The second term is the *function space discrepancy (FSD)*, whose role is to prevent the update from substantially altering the predictions on other data points. The general idea of the FSD term has been successful in alleviating catastrophic forgetting [11], fine-tuning pre-trained models [35], computing influence functions [5], and training a student model from a teacher network [30].

The final term is the *weight space discrepancy (WSD)*, which encourages the update to move the parameters as little as possible. It can be used to motivate damping in the context of second-order optimization [53]. While weight space distance may appear counterproductive from an optimization standpoint because it depends on the model parameterization, analyses of neural network optimization and generalization often rely on network parameters staying close to their initialization in the Euclidean norm [33, 88, 9]. In fact, Wadia et al. [83] showed that pure second-order optimizers (i.e. ones without WSD regularization) are unable to generalize in the overparameterized setting.

Figure 1 illustrates the effects of the WSD and FSD terms on the exact PPM update for a 1D regression example with a batch size of 1. If the proximal objective includes only the loss and WSD term (i.e. $\lambda_{\text{FSD}} = 0$), the PPM update makes the minimal change to the weights which fits the current example, resulting in a global adjustment to the function which overwrites all the other predictions. If only the loss and FSD terms are used (i.e. $\lambda_{\text{WSD}} = 0$), the update carves a spike around the current data point, failing to improve predictions on nearby examples and running the risk of memorization. When both WSD and FSD are penalized, it makes a local adjustment to the predictions, but one which nonetheless improves performance on nearby examples.

## 3.2 Connection Between Proximal Optimization and Second-Order Optimization

We further motivate our proximal objective by relating it to existing neural network optimizers. Ordinary SGD can be viewed as an approximate PPM update with a first-order approximation to the loss term and no FSD term. Hessian-Free optimization [51], a classic second-order optimization method for neural networks, approximately minimizes a second-order approximation to the loss on each batch using conjugate gradients. It can be seen as minimizing a quadratic approximation to Eq. 3 with no FSD term.

| Method | Loss Approx. | FSD | WSD |
|---|---|---|---|
| **Gradient Descent** | 1st-order | - | ✓ |
| **Hessian-Free** | 2nd-order | - | ✓ |
| **Natural Gradient** | 1st-order | 2nd-order | ✗ |
| **Proximal Point** | Exact | Exact | ✓ |

Table 1: Classical 1st and 2nd optimization algorithms interpreted as minimizing approximations of the proximal objective in Eq. 3, using 1st or 2nd order Taylor expansions of the loss or FSD terms.

Amari [1] motivated natural gradient descent (NGD) as a steepest descent method with an infinitesimal step size; this justifies a first-order approximation to the loss term and a second-order approximation to the proximity term. Optimizing over a manifold of probability distributions with KL divergence as the proximity term yields the familiar update involving the Fisher information matrix. Natural gradient optimization of neural networks [52, 54] can be interpreted as minimizing Eq. 3 with a linear approximation to the loss term and a quadratic approximation to the FSD term. While NGD traditionally omits the WSD term in order to achieve parameterization invariance, it is typically included in practical neural network optimizers for stability [54].

In a more general context, when taking a first-order approximation to the loss and a second-order approximation to the FSD, the update rule is given in closed form as:

$$\boldsymbol{\theta}^{(t+1)} \approx \boldsymbol{\theta}^{(t)} - (\lambda_{\text{FSD}}\mathbf{G} + \lambda_{\text{WSD}}\mathbf{I})^{-1}\nabla_{\boldsymbol{\theta}}\mathcal{J}_{\mathcal{B}}(\boldsymbol{\theta}^{(t)}), \tag{4}$$

where $\mathbf{G}$ is the Hessian of the FSD term. The derivation is shown in Appendix F. All of these relationships are summarized in Table 1, and derivations of all of these claims are given in Appendix G.

## 4  Amortized Proximal Optimization

In this section, we introduce Amortized Proximal Optimization (APO), an approach for online meta-learning of optimization parameters. Then, we describe two use cases that we explore in the paper: (1) adapting learning rates of existing base optimizers such as SGD, RMSProp, and Adam, and (2) meta-learning a structured preconditioning matrix.

### 4.1  Proximal Meta-Optimization

We assume an update rule $u$ parameterized by a vector $\phi$ which updates the network weights $\boldsymbol{\theta}$ on a batch $\mathcal{B}^{(t)}$:[2]

$$\boldsymbol{\theta}^{(t+1)} \leftarrow u(\boldsymbol{\theta}^{(t)}, \phi, \mathcal{B}^{(t)}). \tag{5}$$

One use case of APO is to tune the hyperparameters of an existing optimizer, in which case $\phi$ denotes the hyperparameters. For example, when tuning the SGD learning rate, we have $\phi = \eta$ and the update is given by:

$$u_{\text{SGD}}(\boldsymbol{\theta}, \eta, \mathcal{B}) = \boldsymbol{\theta} - \eta\nabla_{\boldsymbol{\theta}}\mathcal{J}_{\mathcal{B}}(\boldsymbol{\theta}). \tag{6}$$

More ambitiously, we could use APO to adapt a full preconditioning matrix $\mathbf{P}$. In this case, we define $\phi = \mathbf{P}$ and the update is given by:

$$u_{\text{Precond}}(\boldsymbol{\theta}, \mathbf{P}, \mathcal{B}) = \boldsymbol{\theta} - \mathbf{P}\nabla_{\boldsymbol{\theta}}\mathcal{J}_{\mathcal{B}}(\boldsymbol{\theta}). \tag{7}$$

In Section 3, we introduced a general proximal objective for training neural networks and observed that many optimization techniques could be seen as an approximation of PPM. Motivated by this connection, we propose to directly minimize the proximal objective with respect to the optimization parameters. While still being able to take advantage of valuable properties of existing optimizers, direct minimization can be robust to cases when the assumptions (such as linear and quadratic approximation of the cost) do not hold. Another advantage of adapting a parametric update rule is that we can amortize the cost of minimizing the proximal objective throughout training.

We propose to use the following meta-objective, which evaluates the proximal objective at $u(\boldsymbol{\theta}, \phi, \mathcal{B})$:

$$\mathcal{Q}(\phi) = \mathbb{E}_{\mathcal{B}\sim\mathcal{D}}\Big[\mathcal{J}_{\mathcal{B}}(u(\boldsymbol{\theta}, \phi, \mathcal{B})) + \lambda_{\text{FSD}}\mathbb{E}_{(\tilde{\mathbf{x}},\cdot)\sim\mathcal{D}}[D_{\text{F}}(u(\boldsymbol{\theta}, \phi, \mathcal{B}), \boldsymbol{\theta}, \tilde{\mathbf{x}})] \tag{8}$$

$$+ \frac{\lambda_{\text{WSD}}}{2}\|u(\boldsymbol{\theta}, \phi, \mathcal{B}) - \boldsymbol{\theta}\|^2\Big].$$

In practice, we estimate the expectations in the meta-objective by sampling two different mini-batches, $\mathcal{B}$ and $\mathcal{B}'$, where $\mathcal{B}$ is used to compute the gradient and the loss term, and $\mathcal{B}'$ is used to compute the FSD term. Intuitively, this proximal meta-objective aims to find optimizer parameters $\phi$ that

---

[2]The update rule may also depend on state maintained by the optimizer, such as the second moments in RMSprop [78]. This state is treated as fixed by APO, so we suppress it to avoid clutter.

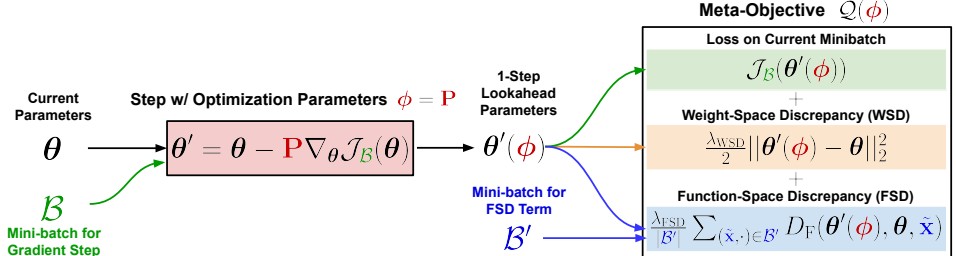

Figure 2: **Amortized Proximal Optimization (APO)**. In each meta-optimization step, we perform a one-step lookahead from the current parameters $\boldsymbol{\theta}$ to obtain updated parameters $\boldsymbol{\theta}'(\boldsymbol{\phi})$, where $\boldsymbol{\phi}$ denotes the optimization parameters (e.g. learning rate $\eta$ or preconditioner $\mathbf{P}$). The meta-objective $\mathcal{Q}(\boldsymbol{\phi})$ then evaluates the proximal point objective at $\boldsymbol{\theta}'(\boldsymbol{\phi})$. Note that the loss term in $\mathcal{Q}(\boldsymbol{\phi})$ is computed on the same data that was used to compute the gradient for the lookahead step, $\mathcal{B}$, while the FSD term is computed using a different datapoint $(\tilde{\mathbf{x}}, \tilde{\mathbf{t}}) \sim \mathcal{D}_{\text{train}}$. The optimization parameters $\boldsymbol{\phi}$ are updated via the meta-gradient $\nabla_{\boldsymbol{\phi}} \mathcal{Q}(\boldsymbol{\phi})$.

---

**Algorithm 1** Amortized Proximal Optimization (APO) — Meta-Learning Optimization Parameters $\boldsymbol{\phi}$

---

**Require:** $\boldsymbol{\theta}$ (initial model parameters), $\boldsymbol{\phi}$ (initial optimization parameters), $K$ (meta-update interval), $\alpha$ (meta-LR)
**Require:** $\lambda_{\text{WSD}}$ (weight-space discrepancy term weighting), $\lambda_{\text{FSD}}$ (function-space discrepancy term weighting)
**while** not converged, iteration $t$ **do**
    $\mathcal{B} \sim \mathcal{D}_{\text{train}}$         ▷ Sample mini-batch to compute the gradient and loss term
    **if** $t \mod K = 0$ **then**         ▷ Perform meta-update every $K$ iterations
        $\mathcal{B}' \sim \mathcal{D}_{\text{train}}$         ▷ Sample additional mini-batch to compute the FSD term
        $\boldsymbol{\theta}'(\boldsymbol{\phi}) := u(\boldsymbol{\theta}, \boldsymbol{\phi}, \mathcal{B})$         ▷ Compute the 1-step lookahead parameters
        $\mathcal{Q}(\boldsymbol{\phi}) := \mathcal{J}_{\mathcal{B}}\left(\boldsymbol{\theta}'(\boldsymbol{\phi})\right) + \lambda_{\text{FSD}}/|\mathcal{B}'| \sum_{(\tilde{\mathbf{x}}, \cdot) \in \mathcal{B}'} D_{\text{F}}(\boldsymbol{\theta}'(\boldsymbol{\phi}), \boldsymbol{\theta}, \tilde{\mathbf{x}}) + \lambda_{\text{WSD}}/2 \left\|\boldsymbol{\theta}'(\boldsymbol{\phi}) - \boldsymbol{\theta}\right\|_2^2$
                              ▷ Compute meta-objective
        $\boldsymbol{\phi} \leftarrow \boldsymbol{\phi} - \alpha\nabla_{\boldsymbol{\phi}}\mathcal{Q}(\boldsymbol{\phi})$         ▷ Update optimizer parameters (e.g. LR or preconditioner)
    **end if**
    $\boldsymbol{\theta} \leftarrow u(\boldsymbol{\theta}, \boldsymbol{\phi}, \mathcal{B})$         ▷ Update model parameters
**end while**

---

minimize the loss on the current mini-batch, while constraining the size of the step with the FSD and WSD terms so that it does not overfit the current mini-batch and undo progress that has been made by other mini-batches.

The optimization parameters $\boldsymbol{\phi}$ are optimized with a stochastic gradient-based algorithm (the *meta-optimizer*). The meta-gradient $\nabla_{\boldsymbol{\phi}}\mathcal{Q}(\boldsymbol{\phi})$ can be computed via automatic differentiation through the one-step unrolled computation graph (Figure 2). We refer to our framework as Amortized Proximal Optimization (APO, Algorithm 1).

### 4.2 APO for Learning Rate Adaptation

One use case of APO is to adapt the learning rate of an existing base optimizer such as SGD. To do so, we let $u_{\text{SGD}}(\boldsymbol{\theta}, \eta, \mathcal{B})$ be the 1-step lookahead of parameters and minimize the proximal meta-objective with respect to the learning rate $\eta$. Although adaptive optimizers such as RMSProp and Adam use coordinate-wise learning rates, they still have a global learning rate which is essential to tune. APO can be applied to such global learning rates to find learning rate schedules (that depend on $\lambda_{\text{FSD}}$ or $\lambda_{\text{WSD}}$).

### 4.3 APO for Adaptive Preconditioning

More ambitiously, we can use the APO framework to adapt the preconditioning matrix, allowing the update rule to flexibly represent various second-order optimization updates. We let $u_{\text{Precond}}(\boldsymbol{\theta}, \mathbf{P}, \mathcal{B})$ denote the parameters after 1 preconditioned gradient step and adapt the preconditioning matrix $\mathbf{P}$ according to our framework.

If the assumptions made when deriving the second-order methods (detailed in Section 3.2) hold, then the optimal preconditioning matrix is equivalent to various second-order updates, depending on the choice of the FSD function.

**Theorem 1.** *Consider an approximation $\hat{\mathcal{Q}}(\mathbf{P})$ to the meta-objective (Eq. 8) where the loss term is linearized around the current weights $\boldsymbol{\theta}$ and the FSD term is replaced by its second-order approximation around $\boldsymbol{\theta}$. Denote the gradient on a mini-batch as $\mathbf{g} = \nabla_{\boldsymbol{\theta}} \mathcal{J}_{\mathcal{B}}(\boldsymbol{\theta})$, and assume that the second moment matrix $\mathbb{E}_{\mathcal{B} \sim \mathcal{D}} \left[ \mathbf{g}\mathbf{g}^\top \right]$ is non-singular. Then, the preconditioning matrix which minimizes $\hat{\mathcal{Q}}$ is given by $\mathbf{P}^\star = (\lambda_{FSD}\mathbf{G} + \lambda_{WSD}\mathbf{I})^{-1}$, where $\mathbf{G}$ denotes the Hessian of the FSD evaluated at $\boldsymbol{\theta}$.*

The proof is provided in Appendix H. As an important special case, when the FSD term is derived from the KL divergence between distributions in output space, $\mathbf{G}$ coincides with the Fisher information matrix $\mathbf{F} = \mathbb{E}_{\mathbf{x} \sim \mathcal{D}, \mathbf{y} \sim P_{\mathbf{y}|\mathbf{x}}(\boldsymbol{\theta})} \left[ \nabla_{\boldsymbol{\theta}} \log p(\mathbf{y}|\mathbf{x}, \boldsymbol{\theta}) \nabla_{\boldsymbol{\theta}} \log p(\mathbf{y}|\mathbf{x}, \boldsymbol{\theta})^\top \right]$, where $P_{\mathbf{y}|\mathbf{x}}(\boldsymbol{\theta})$ denotes the model's predictive distribution over $\mathbf{y}$. Therefore, the optimal preconditioner is the damped natural gradient preconditioner, $\mathbf{P}^\star = (\mathbf{F} + \lambda_{WSD}\mathbf{I})^{-1}$ when $\lambda_{FSD} = 1$. Thus, when APO is used to exactly minimize an approximate meta-objective, the update it yields coincides with classical second-order optimization algorithms, depending on the choice of the FSD function.

## 4.4 Structured Preconditioner Adaptation

In the previous sections, the discussion assumed a full preconditioning matrix for simplicity. However, a full preconditioner is impractical to represent for modern neural networks. Moreover, for practical stability of the learned preconditioned update, we would like to enforce the preconditioner to be positive semidefinite (PSD) so that the transformed gradient is a descent direction [64].

To satisfy these requirements, we adopt a structured preconditioner analogous to that of the EKFAC optimizer [24]. Given a weight matrix $\mathbf{W} \in \mathbb{R}^{m_i \times m_{i+1}}$ of a layer, we construct the preconditioning matrix as a product of smaller matrices:

$$\mathbf{P}_S = (\mathbf{A} \otimes \mathbf{B})\text{diag}(\text{vec}(\mathbf{S}))^2(\mathbf{A} \otimes \mathbf{B})^\top, \tag{9}$$

where $\mathbf{A} \in \mathbb{R}^{m_{i+1} \times m_{i+1}}$, $\mathbf{B} \in \mathbb{R}^{m_i \times m_i}$, and $\mathbf{S} \in \mathbb{R}^{m_i \times m_{i+1}}$ are small block matrices. Here, $\otimes$ denotes the Kronecker product, $\text{diag}(\cdot)$ denotes the diagonalization operator, and $\text{vec}(\cdot)$ denotes the vectorization operator. This parameterization is memory efficient: it requires $m_i^2 + m_{i+1}^2 + m_i m_{i+1}$ parameters to store, as opposed to $m_i^2 m_{i+1}^2$ parameters for a full preconditioning matrix. It is straightforward to show that the structured preconditioner in Eq. 9 is PSD, as it takes the form $\mathbf{C}\mathbf{D}\mathbf{C}^\top$, where $\mathbf{D}$ is PSD. The preconditioned gradient can be computed efficiently by using the properties of the Kronecker product:

$$\mathbf{P}_S\text{vec}(\nabla_{\mathbf{W}} \mathcal{J}_{\mathcal{B}}(\boldsymbol{\theta})) = \text{vec}(\mathbf{B}(\mathbf{S}^2 \odot \mathbf{B}^\top \nabla_{\mathbf{W}} \mathcal{J}_{\mathcal{B}}(\boldsymbol{\theta})\mathbf{A})\mathbf{A}^\top), \tag{10}$$

where $\odot$ denotes elementwise multiplication. This is tractable to compute as it only requires four additional matrix multiplications and elementwise multiplication of small block matrices in each layer when updating the parameters. While EKFAC uses complicated covariance estimation and eigenvalue decomposition to construct the block matrices, in APO, we meta-learn these block matrices directly, where $\boldsymbol{\phi} = [\text{vec}(\mathbf{A})^\top, \text{vec}(\mathbf{B})^\top, \text{vec}(\mathbf{S})^\top]^\top$. As APO does not require inverting (or performing eigendecompositions of) the block matrices, our structured representation incurs less computation per iteration than EKFAC.

While we defined an optimizer with the same functional form as EKFAC, it is not immediately obvious whether the preconditioner which is actually learned by APO will be at all similar. A Corollary of Theorem 1 shows that if certain conditions are satisfied, including the assumptions underlying KFAC [54], then the structured preconditioner minimizing Eq. 8 coincides with KFAC:

**Corollary 2.** *Suppose that (1) the assumptions for Theorem 1 are satisfied, (2) the FSD term measures the KL divergence, and (3) $\lambda_{WSD} = 0$ and $\lambda_{FSD} = 1$. Moreover, suppose that the parameters $\boldsymbol{\theta}$ satisfy the KFAC assumptions listed in Appendix I. Then, the optimal solution to the approximate meta-objective recovers the KFAC update, which can be represented using Eq. 9.*

The proof is in Appendix I. If the KFAC assumptions are not satisfied, then APO will generally learn a different preconditioner. This may be desirable, especially if differing probabilistic assumptions lead to different update rules, as is the case for KFAC applied to convolutional networks [26, 38].

## 4.5 Computation and Memory Cost

**Computation Cost.** Computing the FSD term requires sampling an additional mini-batch from the training set and performing two additional forward passes for $f(\tilde{\mathbf{x}}, \boldsymbol{\theta})$ and $f(\tilde{\mathbf{x}}, u(\boldsymbol{\theta}, \boldsymbol{\phi}, \mathcal{B}))$. Combined with the loss term evaluated on the original mini-batch, one meta-optimization step requires three forward passes to compute the proximal objective. It additionally requires a backward pass through the 1-step unrolled computation graph (Figure 2) to compute the gradient of the proximal meta-objective $\mathcal{Q}(\boldsymbol{\phi})$ with respect to $\boldsymbol{\phi}$. This overhead can be reduced by performing a meta-update only once every $K$ iterations: the overhead will consist of $3/K$ additional forward passes and $1/K$ additional backward passes per iteration, which is small for modest values of $K$ (e.g., $K = 10$).

**Memory Cost.** APO requires twice the model memory for the 1-step unrolling when computing the proximal meta-objective. In the context of structured preconditioner adaptation, we further need to store block matrices $\mathbf{A}$, $\mathbf{B}$, and $\mathbf{S}$ (in Eq. 9) for each layer, as in KFAC and EKFAC.

## 5 Related Work

We provide extended related work and a conceptual comparison of meta-optimization methods in Appendix B and Table 6, respectively.

**Gradient-Based Learning Rate Adaptation.** Maclaurin et al. [50] backpropagate through the full unrolled training procedure to meta-optimize learning rate schedules *offline*. This is expensive, as it requires completing a full training run to make a single hyperparameter update. A related approach is to unroll the optimization for a small number of steps and perform truncated backpropagation [16, 22]. Micaelli et al. [58] perform offline hyperparameter optimization using forward-mode (FDS) rather than reverse-mode gradient accumulation. FDS performs one update to the hyperparameters after each full training run, and thus requires multiple training runs, in contrast to APO, which operates within a single training run. Hypergradient descent [6] adapts the learning rate to minimize the expected loss in the next iteration.

**Second-Order Optimization.** Although preconditioned methods have better convergence rates than first-order methods [8, 64], storing and computing the inverse of preconditioning matrices is impractical for high-dimensional problems. To mitigate these computational issues, Hessian-free optimization [51, 55] approximates Newton's update by only accessing the curvature matrix through Hessian-vector products. Other works impose a structure on the preconditioner by representing it as a Kronecker product [54, 26, 56, 24, 27, 77], a diagonal matrix [19, 36], or a low-rank matrix [39, 60]. However, these approximate second-order methods may not be easy to implement in deep learning frameworks, and can still be expensive as they often require matrix inversion or eigendecomposition.

**Gradient-Based Preconditioner Adaptation.** There has been some prior work on meta-learning preconditioners. Moskovitz et al. [61] learn the preconditioning matrix with hypergradient descent. Meta-curvature [66] and warped gradient descent [41, 21] adapt the preconditioning matrix that yields effective parameter updates across diverse tasks in the context of few-shot learning.

## 6 Experiments

Our experiments investigate the following questions: (1) How does the structured preconditioning matrix adapted by APO perform in comparison to existing first- and second-order optimizers? (2) How does the learning rate adapted by APO perform compared to optimal fixed learning rates and manual decay schedules commonly used in the literature?

We used APO to meta-learn the preconditioning matrices for a broad range of tasks, including several regression datasets, autoencoder training, image classification on CIFAR-10 and CIFAR-100 using several network architectures, neural machine translation using transformers, and low-precision (16-bit) training. Several of these tasks are particularly challenging for first-order optimizers. In addition, we used APO to tune the global learning rate for multiple base optimizers – SGD, SGD with momentum (denoted SGDm), RMSprop, and Adam – on CIFAR-10 and CIFAR-100 classification with several network architectures.

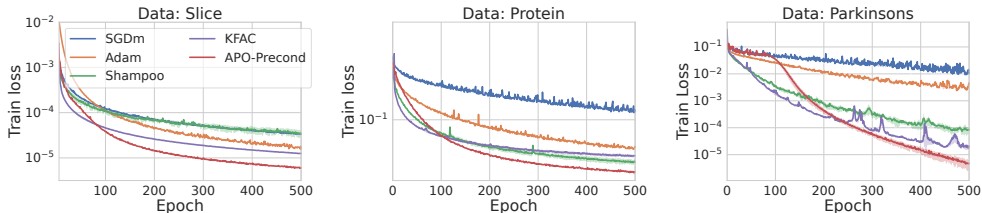

Figure 3: A comparison of SGDm, Adam, KFAC, Shampoo, and APO-Precond on UCI regression tasks. Across all tasks, APO-Precond achieves lower loss with competitive convergence compared to second-order optimizers.

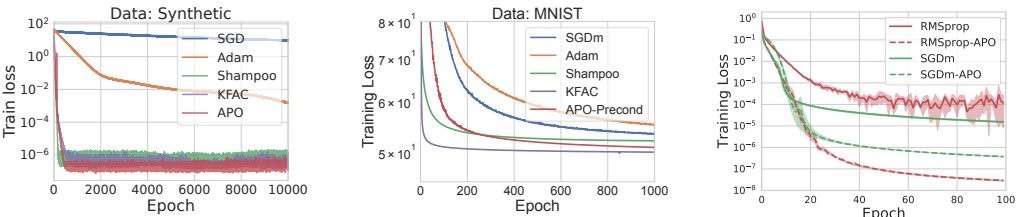

Figure 4: **Left:** Synthetic data for poorly-conditioned regression; **Middle:** Deep autoencoder on MNIST; **Right:** Tuning the global learning rate with APO—we show the training loss for a two-layer MLP trained on MNIST, using SGDm and RMSprop (solid lines), and their APO-tuned variants (dashed lines).

We denote our method that adapts the structured preconditioning matrix as "APO-Precond". The method that tunes the global learning rate of a base optimizer is denoted as "Base-APO" (e.g. SGDm-APO). Experiment details and additional experiments, including ablation studies, are given in Appendix C and J, respectively.

## 6.1 Meta-Learning Preconditioners

**Poorly-Conditioned Regression.** We fist considered a regression task traditionally used to illustrate failures of neural network optimization [68]. The targets are given by $\mathbf{t} = \mathbf{A}\mathbf{x}$, where $\mathbf{A}$ is an ill-conditioned matrix with $\kappa(\mathbf{A}) = 10^{10}$. We trained a two-layer linear network $f(\mathbf{x}, \boldsymbol{\theta}) = \mathbf{W}_2\mathbf{W}_1\mathbf{x}$ and minimized the objective $\mathcal{J}(\boldsymbol{\theta}) = \mathbb{E}_{\mathbf{x} \sim \mathcal{N}(\mathbf{0}, \mathbf{I})} \left[ \|\mathbf{A}\mathbf{x} - \mathbf{W}_2\mathbf{W}_1\mathbf{x}\|^2 \right]$. In Figure 4 (left), we show training curves for SGDm, Adam, Shampoo [27], KFAC, and APO-Precond. As the problem is ill-conditioned, 2nd-order optimizers such as Shampoo and KFAC converge faster than 1st-order methods. APO-Precond performs comparably to 2nd-order optimizers with lower loss than KFAC.

**UCI Regression.** Next, we validated APO-Precond on the Slice, Protein, and Parkinsons datasets from the UCI regression collection [18]. We trained a 2-layer MLP with 100 hidden units per layer and ReLU activations for 500 epochs. The training curves for each optimizer are shown in Figure 3. By tuning the preconditioning matrix during training, APO-Precond consistently achieved competitive convergence compared to other second-order optimizers and reached lower training loss than all baselines.

**Image Reconstruction.** We trained an 8-layer autoencoder on MNIST [40]; this is known to be a challenging optimization task for first-order optimizers [31, 55, 54]. We followed the experimental set-up from Martens & Grosse [54], where the encoder and decoder consist of 4 fully-connected layers with sigmoid activation. The decoder structure is symmetric to that of the encoder, and they do not have tied weights. The logistic activation function and the presence of a bottleneck layer make this a challenging optimization problem compared with most current-day architectures. We compare APO-Precond with SGDm, Adam, Shampoo, and KFAC optimizers and show the training losses for each optimizer in Figure 4 (middle). APO-Precond converges faster than first-order methods and achieves competitive training loss to other second-order methods (although there remains a performance gap compared with KFAC).

---

[3]We used AdamW optimizer [48] for training Transformer model.

| Task | Model | SGDm | Adam | KFAC | APO-Precond |
|------|-------|------|------|------|-------------|
| **CIFAR-10** | **LeNet** | 75.73 | 73.41 | 76.63 | **77.42** |
| **CIFAR-10** | **AlexNet** | 76.27 | 76.09 | 78.33 | **81.14** |
| **CIFAR-10** | **VGG16** | 91.82 | 90.19 | 92.05 | **92.13** |
| **CIFAR-10** | **ResNet-18** | 93.69 | 93.27 | 94.60 | **94.75** |
| **CIFAR-10** | **ResNet-32** | 94.40 | 93.30 | 94.49 | **94.83** |
| **CIFAR-100** | **AlexNet** | 43.95 | 41.82 | 46.24 | **52.35** |
| **CIFAR-100** | **VGG16** | 65.98 | 60.61 | 61.84 | **67.95** |
| **CIFAR-100** | **ResNet-18** | 76.85 | 70.87 | 76.48 | **76.88** |
| **CIFAR-100** | **ResNet-32** | **77.47** | 56.58 | 75.70 | 77.41 |
| **SVHN** | **ResNet-18** | 96.19 | 95.59 | 96.08 | **96.89** |
| **IWSLT14** | **Transformer** | 31.43 | 34.60 [3] | - | **34.62** |

Table 2: Test accuracy on CIFAR-10 and CIFAR-100, and BLEU score on the IWSLT'14 German-to-English translation dataset for various optimizers.

**Image Classification.** To investigate whether adapting the preconditioner with APO improves generalization, we conducted image classification experiments on CIFAR-10 and CIFAR-100. We trained LeNet [40], AlexNet [37], VGG-16 [71] (w/o batch norm [32]), ResNet-18, and ResNet-32 [29] architectures for 200 epochs on batches of 128 images. The test accuracies for SGDm, Adam, KFAC, and APO-Precond are shown in Table 2. We found that APO-Precond achieved competitive generalization performance to SGDm and KFAC. In particular, for architectures without batch normalization (LeNet, AlexNet, and VGG-16), APO-Precond improved the test accuracy substantially.

**Neural Machine Translation.** To verify the effectiveness of APO on various tasks, we applied APO-Precond on the IWSLT'14 German-to-English translation task [15]. We used a Transformer [79] composed of 6 encoder and decoder layers, with word embedding and hidden vector dimensionality 512. We compared APO-Precond to SGDm and AdamW [48]. For AdamW, we used a warmup-then-decay learning rate schedule widely used in practice, and for SGD and APO-Precond, we kept the learning rate fixed after the warmup. In Table 2, we show the final test BLEU score for SGDm, AdamW, and APO-Precond. While keeping the learning rate fixed, we achieved a BLEU score competitive with AdamW.

**Low Precision Training.** Low precision training presents a challenge for second-order optimizers such as KFAC which rely on matrix inverses that may be sensitive to quantization noise. We trained LeNet and ResNet-18 with 16-bit floating-point arithmetic to examine if APO-Precond is applicable in training the net-

| Task | Model | SGDm | KFAC | APO-P |
|------|-------|------|------|-------|
| **CIFAR-10** | **LeNet** | 75.65 | 74.95 | **77.25** |
| **CIFAR-10** | **ResNet-18** | 94.15 | 92.72 | **94.79** |
| **CIFAR-100** | **ResNet-18** | 73.53 | 73.12 | **75.47** |

Table 3: Test accuracy of 16-bit networks on CIFAR-10 and CIFAR-100.

works in lower precision. We used the experimental setup from Section 6.1 but stored parameters, activations, and gradients in 16-bit precision. We found that KFAC required a large damping factor to maintain stability, and this prevented it from fully utilizing curvature information. In contrast, as APO-Precond does not require matrix inversion, it remained stable with the same choice of FSD and WSD weights we used in the full precision experiments. The final test accuracies on ResNet-18 for SGDm, KFAC, and APO-Precond are shown in Table 3.

## 6.2 Meta-Learning Learning Rates

**Image Classification on MNIST.** First, we compared SGDm and RMSprop to their APO-tuned variants to train an MLP on MNIST. We used a two-layer MLP with 1000 hidden units per layer and ReLU nonlinearities, and trained on mini-batches of size 100 for 100 epochs. Figure 4 (Right) shows the training loss achieved by each approach; we found that for both base optimizers, APO improved convergence speed and obtained substantially lower loss than the baselines.

**Image Classification on CIFAR-10 & CIFAR-100.** For learning rate adaptation on CIFAR-10, we experimented with three network architectures: ResNet32 [29], ResNet34, and WideResNet

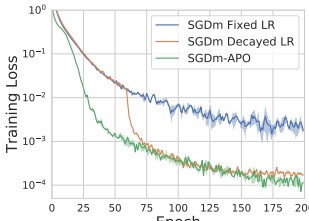 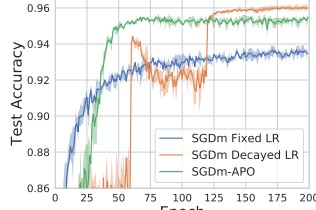 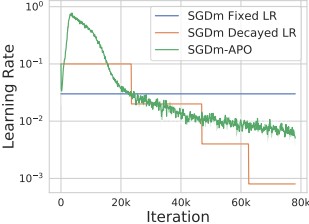

Figure 5: WideResNet 28-10 on CIFAR-10, using SGD with momentum (SGDm). We compare the training loss (**left**), test accuracy (**middle**), and learning rate schedules (**right**) of baselines with fixed and decayed learning rates, and APO. The training loss plot uses hyperparameters chosen based on training loss, while the middle and right plots use hyperparameters chosen based on validation accuracy. The shaded regions show the min/max values over 4 random restarts.

|  | **C-10, ResNet-32** | | | **C-10, ResNet-34** | | | **C-10, WRN 28-10** | | | **C-100, WRN 28-10** | | |
|---|---|---|---|---|---|---|---|---|---|---|---|---|
|  | **Fixed** | **Decay** | **APO** | **Fixed** | **Decay** | **APO** | **Fixed** | **Decay** | **APO** | **Fixed** | **Decay** | **APO** |
| **SGD** | 90.07 | 93.30 | 92.71 | 93.00 | 93.54 | 94.27 | 93.38 | 94.86 | 94.85 | 76.29 | 77.92 | 76.87 |
| **SGDm** | 89.40 | 93.34 | 92.75 | 92.99 | 95.08 | 94.47 | 93.46 | 95.98 | 95.50 | 74.81 | 81.01 | 79.33 |
| **RMSprop** | 89.84 | 91.94 | 91.28 | 92.87 | 93.87 | 93.97 | 92.91 | 93.60 | 94.22 | 72.06 | 76.06 | 74.17 |
| **Adam** | 90.45 | 92.26 | 91.81 | 93.23 | 94.12 | 93.80 | 92.81 | 94.04 | 93.83 | 72.01 | 75.53 | 76.33 |

Table 4: Tuning the global LR for CIFAR-10 ("C-10") and CIFAR-100 ("C-100"): We compare the test accuracies achieved by the optimal fixed learning rate, the manual step decay schedule, and the APO-adapted schedule, using ResNet-32 [29], ResNet-34, and WideResNet 28-10 [86]. Results are the mean of four random restarts. APO outperforms optimal fixed learning rates, and is often competitive with manual schedules. APO generally achieves test accuracy comparable to manual schedules in fewer training iterations (App. D).

(WRN-28-10) [86]. For ResNet32, we trained for 400 epochs, and the decayed baseline used a step schedule with $10\times$ decay at epochs 150 and 250, following [49]. For ResNet34 and WRN-28-10, we trained for 200 epochs, and the decayed baseline used a step schedule with $5\times$ decay at epochs 60, 120, and 160, following [86]. For CIFAR-100, we used WRN-28-10 with the same schedule as for CIFAR-10. For each of the base optimizers, we compared APO to (1) the optimal fixed learning rate and (2) a manual step learning rate decay schedule. The test accuracies for each base optimizer and their APO-tuned variants are shown in Table 4. In addition, Figure 5 shows the training loss, test accuracy and learning rate adaptation for WRN-28-10 on CIFAR-10, using SGDm as the base optimizer. Using APO to tune the global learning rate yields higher test accuracy than the best fixed learning rate, and is competitive with the manual schedule.

## 7 Conclusion

We introduced Amortized Proximal Optimization (APO), a framework for online meta-learning of optimization parameters which approximates the proximal point method by learning a parametric update rule. As the meta-parameters are updated only once per $K$ steps of optimization, APO incurs minimal computational overhead. We applied APO to two settings: (1) meta-learning the global learning rate for existing base optimizers (e.g., SGD, RMSprop, and Adam) and (2) meta-learning structured preconditioning matrices, which provides a new approach to 2nd-order optimization. Compared to methods such as KFAC, APO eliminates the need to compute matrix inverses, yielding improved efficiency and numerical stability. On a range of tasks, we showed that APO is competitive with 2nd-order methods, and improves generalization compared to baseline 1st- and 2nd-order optimizers.

## Acknowledgements

We thank Michael Zhang for valuable feedback on this paper and thank Alston Lo for helping setting up the experiments. We would also like to thank Saminul Haque, Jonathan Lorraine, Denny Wu, and Guodong Zhang, and our many other colleagues for their helpful discussions throughout this research. Resources used in this research were provided, in part, by the Province of Ontario, the Government of Canada through CIFAR, and companies sponsoring the Vector Institute (`www.vectorinstitute.ai/partners`).

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
