# OpenReview forum: "Amortized Proximal Optimization"
_NeurIPS.cc/2022/Conference — NeurIPS 2022 Accept_

### Official Review · Reviewer_nHHL · 2022-07-11

**Rating:** 7
**Confidence:** 3
**Soundness:** 2 fair
**Presentation:** 3 good
**Contribution:** 2 fair

**Summary:**

This paper introduces an online meta-optimization method called Amortized Proximal Optimization (APO) that can dynamically adjusts parameters like learning rate and preconditioning matrices during the training process.  Before introducing APO, background information is provided to motivate the necessity for both  a function space distance (FSD) term and a weight space distance (WSD) term for the proximal objective. Then, prior 1st and 2nd order optimization methods are interpreted as proximal point methods with various approximations for the loss, FSD, and WSD.  APO is proposed as a meta-optimizer that can be combined with existing optimization strategies to dynamically adjust meta-parameters.  Experiments show APO adaptively learns good learning rate schedules that outperform fixed learning rates and are competitive with manually defined learning rate schedules.  When applied to preconditioning matrices, APO achieves optimization performance competitive with more computationally expensive second order approaches.

**Questions:**

- Is there any intuition for the empirical results in Section E of the Appendix where the same minibatch needs to be used to compute the the gradient of the base optimizer and the loss term of the meta-objective while a different minibatch should be used for the FSD term?
- How should APO be extended to adapt multiple meta-parameters? e.g. both learning rate and preconditioning matrix?  Are there meta-parameters that are not compatible with APO? can momentum be adapted with APO?  What about generalization/regularization parameters like weight-decay?
- It seems like manual learning rate schedules outperfrom APO-lr due to using smaller learning rates towards the end of training.  Why does APO not learn to decay LR even more/faster?  Should the \lambda parameters of APO be adjusted during training as well to promote faster convergence?
- I suggest including a legend for Figure 4 left.
- See additional suggestions/questions in Quality section of the box above.

__Typos__
- Footnote 2 is missing.
- ln 289, extra "a"

**Limitations:**

The authors do not discuss the consistent underperformance of APO relative to using a manual decay schedule for many vision tasks in Table 4.  I think it is worth understanding why there's this gap and see if it's possible to bridge this gap with modifications to APO.  I also think a discussion of potential additional compute cost associated with tuning \lambda for APO would be useful.  Finally, the authors make it sound like APO is applicable to arbitrary meta-parameters but the questions above regarding optimizing multiple meta-parameters simultaneously and regularization related meta-parameters are unanswered.

**Strengths And Weaknesses:**

Strengths:
- Well written paper that provides useful insights into connections between proximal optimization and popular optimization methods and the role of FSD and WSD in improving performance.
- Extensive empirical results across multiple types of problems and architectures.
- Strong results for APO-Precond that show improved performance for poorly conditioned regression problems and FP16 training.

Weaknesses:
- Presentation of experimental results can be improved.
- APO underperforms SGDm with manual decay schedule in nearly all tasks in table 4.
- APO in some sense trades off selection a learning rate and decay schedule with tuning \lambda for the proximal penalty terms.  Perhaps this is not surprising since we shouldn't expect a free lunch and a more powerful optimization should require more tuning.
- The decision to use a separate batch for the FSD term of the meta-objective feels heuristic and is only supported by empirical evidence in Appendix E.


## Originality
I am not too familiar with work on developing new optimization methods but I found the presentation and methodology to be interesting and original.

## Quality
The overall quality of the paper is high with adequate detail provided in the supplementary material for all experiments.  I do think the experimental results need to be improved however in the following ways:
- Error bars/std need to be provided for Table 2 and are also highly recommended for all other tables.  For Table 2 in particular, the differences between KFAC and APO-P on architectures with batch-norm layers seem very small and likely insignificant.
- Table 4 can use bolding/underlines for users to quickly identify the best optimizer in each setting.  I think this will show that manual decay schedule does the best in nearly every setting.  While this result detracts from the benefit of APO, I think it's worth investigating how to close this gap.
- What is the computational cost for KFAC and APO-P in Section 5.1?

## Clarity
The paper is well written in most places.  I do have a few suggestion/questions:
- VGG-16 is stated to not have batch-norm in ln 286-287 but then in ln 296, only LeNet and AlexNet are considered to not have batch-norm.
- ln 32-34 can be worded better.
- ln 142-145: This part was confusing for me and I could have used a concrete example.

## Significance
I think this work can potentially have high significance if the authors are able to close the gap between manual decay and learning lr with APO and show how APO can be applied to multiple meta-parameters.  It still has impact in the current form

---

> ### Author Response · Authors · 2022-08-02
> **Response to Reviewer nHHL (Part 1)**
>
> **Q: Is there any intuition for the empirical results in Section E of the Appendix where the same minibatch needs to be used to compute the gradient of the base optimizer and the loss term of the meta-objective while a different minibatch should be used for the FSD term?**
>
> Using a separate mini-batch is motivated directly from the definition of the function-space discrepancy as an expectation over the data generating distribution. For computational efficiency, we use a single-sample Monte-Carlo estimate of this expectation rather than computing it over the full training set. Using the same minibatch to compute the FSD term would result in a different meta-objective; our meta-objective encourages the model to remember what it learned on other mini-batches, e.g., enforcing the model’s predictions on randomly sampled batch during training to be consistent before and after a parameter update.
> Using a random minibatch to evaluate the loss after taking a lookahead step corresponds to the meta-objective analyzed in [1] (which we discuss in Appendix B (“Extended Related Work”)). This meta-objective was shown to yield undesirable behavior, where the learning rate decreases rapidly to reduce fluctuations of the loss caused by stochastic mini-batch evaluation. Evaluating the lookahead step on the same minibatch that was used to compute the gradient for the step avoids this issue, and leads to larger learning rates; the function-space and weight-space discrepancy terms regularize the step such that it does not move too far, e.g., decreasing the loss on the current mini-batch but altering predictions on other mini-batches. Moreover, using the same mini-batch is essential for the proof of Theorem 1; the use of the same mini-batch and assuming the linearized loss term and quadratic FSD term allows our meta-objective to find a preconditioner that recovers many existing second-order optimization methods, $\mathbf{P}^{\star} = (\lambda_{\text{FSD}} \mathbf{G} + \lambda_{\text{WSD}} \mathbf{I})^{-1}$.
>
>
> **Q: How should APO be extended to adapt multiple meta-parameters? Are there meta-parameters that are not compatible with APO? Can momentum be adapted with APO? What about generalization/regularization parameters like weight decay?**
>
> APO is a general framework for adapting optimization parameters. While our paper demonstrates two specific use cases (tuning global learning rate or adapting the preconditioning matrix), in principle, APO can be used to adapt other optimization parameters (e.g., the damping coefficient in KFAC) simultaneously. For tuning multiple optimization parameters, one simply needs to perform gradient descent on these parameters simultaneously (just as we take gradient with respect to the learning rate or preconditioning matrix).
>
> Note that APO is not designed to tune regularization hyperparameters like $L_2$ regularization, data augmentation, dropout, etc. Since adapting the momentum coefficient requires considering long-term performance, it is not appropriate to adapt it with a one-step meta-objective like the one used in APO. Instead, we can adapt the meta-parameters with APO as if there is no momentum but then apply a fixed momentum coefficient (e.g., 0.9) on top of the updates.
>
> We have added a discussion to clarify the scope of APO to Section B (“Extended Related Work”).
>
> **Q: I suggest including a legend for Figure 4 left.**
>
> Thank you for the suggestion; we have included this legend in the updated paper.

---

> > ### Author Response · Authors · 2022-08-02
> > **Response to Reviewer nHHL (Part 2)**
> >
> > **Q: APO in some sense trades off selecting a learning rate and decay schedule with tuning $\lambda$ for the proximal penalty terms. Perhaps this is not surprising since we shouldn't expect a free lunch and a more powerful optimization should require more tuning.**
> >
> > The reviewer is correct in that APO requires tuning of $\lambda$. When tuning a global learning rate (APO-LR), we trade-off choosing a fixed value of $\lambda$ for choosing a learning rate schedule. When tuning the preconditioner (APO-Precond), we trade-off choosing a value of $\lambda$ for tuning a high-dimensional block-diagonal preconditioner, which would be impossible to tune by hand (and challenging to tune even with existing gradient-based methods).
> >
> > The detailed procedure for selecting the proximal coefficients is described in Appendix C (“Experimental Details”). The search space for $\lambda$ configuration was generally similar to the learning rate search space size for first-order baseline optimizers. For example, in the image classification task (ResNet-18 & CIFAR-10), APO-Precond searched over 8 different $\lambda$ configurations, and SGDm searched over 8 different learning rate configurations. We also include ablation studies over these hyperparameters in Appendix J (“Ablations”) and show that APO is not highly sensitive to $\lambda_{\text{FSD}}$ and $\lambda_{\text{WSD}}$. Hence, in practice, APO does not require an intensive search over proximal coefficients to obtain better convergence and performance than other second-order methods.
> >
> > An interesting finding is that APO chooses sensible schedules, rather than getting stuck due to short horizon bias like other meta-optimization methods.
> >
> >
> > **Q: Error bars/std need to be provided for Table 2 and are also highly recommended for all other tables. For Table 2 in particular, the differences between KFAC and APO-P on architectures with batch-norm layers seem very small and likely insignificant.**
> >
> > In the updated version, we have added expanded tables with standard deviations in Appendix D (due to a lack of space to add standard deviations in the main text). Overall, we found the results to be consistent between random initializations, both for baselines and for APO.
> > We would like to note that ResNet baselines on the CIFAR dataset generally do not have much room for significant improvement. We believe that a consistent 0.15-1.50 percent improvement in accuracy over KFAC demonstrates the effectiveness of APO-Precond.
> >
> >
> > **Q: Table 4 can use bolding/underlines for users to quickly identify the best optimizer in each setting.**
> >
> > We have added background colors in Table 4 to make the comparison clearer, where red denotes lower accuracy and green denotes higher accuracy. You are correct that the manual decay schedule typically yields the best performance, but APO achieves performance closer to the manual schedule than the fixed learning rate runs. In addition, we have added comparisons to two other methods for online learning rate adaptation, Hypergradient Descent and L4, in Appendix D.6; empirically, both of these methods obtained lower test accuracy than APO, such that APO was the closest in performance to the manual decay schedule.
> >
> > **Q: What is the computational cost for KFAC and APO-P in Section 5.1?**
> >
> > As detailed in Section 3.5, APO requires 3/K additional forward passes and 1/K additional backward passes for each base parameter update, where K is the interval of meta-update. We set K=10 in all our experiments. In Appendix J.2, we also show that APO is robust to the values of K, and hence K can be set higher to further reduce the computational overhead.
> > KFAC requires computing the gradient covariance in each iteration and inverting block matrices in every M iteration, where M=100 following [2]. Hence, it is difficult to directly compare the computational cost between KFAC and APO-P. However, in Figure 10 (Appendix D.3), we show the training curve as a function of wall-clock time on CIFAR-10 & AlexNet. Using our default hyperparameter K=10, APO-Precond terminates training faster while obtaining better results than KFAC.

---

> > > ### Author Response · Authors · 2022-08-02
> > > **Response to Reviewer nHHL (Part 3)**
> > >
> > > **Q: Why does APO not learn to decay LR even more/faster? Should the $\lambda$ parameters of APO be adjusted during training as well to promote faster convergence?**
> > >
> > > You are correct that adapting $\lambda$ may lead to smaller learning rates at the end of the training, which may benefit final performance. Considering automated methods for tuning $\lambda$ gives rise to meta-meta learning algorithms, which we leave for future work.
> > >
> > >
> > > **Q: The authors do not discuss the consistent underperformance of APO relative to using a manual decay schedule for many vision tasks in Table 4.**
> > >
> > > In our manuscript, we claim that “APO outperforms the best fixed learning rates, and is competitive with manual decay schedules.” However, empirically APO obtains better performance than the other LR adaptation techniques we compared to, Hypergradient Descent and L4, in Appendix D.6.
> > >
> > > | Method      | Fixed | Decayed |   HD  |   L4  | APO (Ours) |
> > > |-------------|:-----:|:-------:|:-----:|:-----:|:----------:|
> > > | **SGD**     | 93.00 |  93.54  | 92.53 | 85.93 |    94.27   |
> > > | **SGDm**    | 92.99 |  95.08  | 92.79 | 88.35 |    94.47   |
> > > | **RMSprop** | 92.87 |  93.87  |   -   | 80.93 |    93.97   |
> > > | **Adam**    | 93.23 |  94.12  | 92.54 | 85.10 |    93.80   |
> > >
> > >
> > > **Q: VGG-16 is stated to not have batch-norm in ln 286-287 but then in 296, only LeNet and AlexNet are considered to not have batch-norm.**
> > >
> > > Thank you for your careful reading, we fixed line 296 to state that LeNet, AlexNet, and VGG-16 do not have batch-norm.
> > >
> > >
> > > **Q: ln 32-34 can be worded better.**
> > >
> > > Thank you for the suggestion. We have re-worded this sentence in the updated version to be: “Minimization of the exact proximal objective in Eq. 1 is un-economical, as it may require tens to hundreds of optimization steps per parameter update. Instead, we propose to learn a parametric update rule $\boldsymbol{\theta}^{(t+1)} \gets u(\boldsymbol{\theta}^{(t)}, \mathcal{B}^{(t)}, \boldsymbol{\phi})$ described by meta-parameters $\boldsymbol{\phi}$, and use a variant of Eq. 1 as a _meta-objective_ to optimize $\boldsymbol{\phi}$. By meta-learning the optimization parameters online, we can amortize the cost of minimizing the PPM objective, by performing one meta-step every $K > 1$ steps of the base optimization.”
> > >
> > >
> > > **Q: ln 142-145: This part was confusing for me and I could have used a concrete example.**
> > >
> > > Apologies for the confusion, we have updated the writing for this part as follows: “As shown in Table 1, under linear and quadratic assumptions on the loss term and FSD term, the proximal objective in Eq. 1 has a closed-form solution; however, when these assumptions do not hold, direct minimization of the proximal objective can still be robust. Adapting a parametric update rule allows us to take advantage of the properties of existing optimizers, and also amortizes the cost of minimizing the proximal objective over the course of training.”
> > >
> > >
> > > [1] Wu et al., “Understanding Short-Horizon Bias in Stochastic Meta-Optimization,” ICLR 2018.
> > >
> > > [2] Zhang et al., “Three Mechanisms of Weight Decay Regularization,” ICLR 2019.

---

### Official Review · Reviewer_NLDo · 2022-07-12

**Rating:** 7
**Confidence:** 5
**Ethics Flag:** Yes
**Soundness:** 3 good
**Presentation:** 4 excellent
**Contribution:** 3 good

**Summary:**

This paper proposes Amortized Proximal Optimization (APO) which is an online meta-optimization for optimization parameters. The modern neural network update rule can be interpreted by proximal point update where it find minimization of three different function w.r.t update rule: 1) the current mini-batch loss, 2) function discrepancy, and 3) weight discrepancy. The authors also reveal that this interpretation is related to other update rule such as Gradient Descent, Hessian-Free, and Natural Gradient. Based on this interpretation, they propose a new meta-objective by sampling two different mini-batches when updating meta-parameters. This paper uses the proposed meta-optimization for optimizing learning rate adaptation and preconditioning matrix for the update rule, and the extensive experimental results shows the effectiveness of APO.

**Questions:**

- This paper use an efficient approximation for optimizing the meta-objective, by considering only 1-step horizon.  If one update the meta-parameter for every $K$ step, it is natural for me to unroll all the previous K steps (if it is possible). For example, either FMD (RMD is possible for preconditioning matrix) [3], or implicit differentiation [2] are feasible to unroll them. Can you elaborate the reason why one should use 1-step approximation?

[3]  Luca Franceschi, Michele Donini, Paolo Frasconi, and Massimiliano Pontil. Forward and reverse gradient-based hyperparameter optimization. In International Conference on Machine Learning, pages 1165–1173. PMLR, 2017.

**Ethics Review Area:**

["I don’t know"]

**Limitations:**

I did not find any potential negative societal impact of this paper.

**Strengths And Weaknesses:**

- Originality

Many other approaches that meta-learns optimization parameters focus on minimizing training (or validation) loss w.r.t to the parameters. The main originality of this paper is the proposed meta-objective which includes two different discrepancy term, and it is novel.

---

- Quality

The intuition of this paper is well-motivated throughout the paper, by relating modern neural networks update rule to the proximal point method. Based on the claim on the relatedness, the derivation of the meta-objective seems technically sound to me. The experiments are sufficient for me to understand the effectiveness of the propose method.

---

- Clarity

This paper is well-written (especially the Figure 2 and Algorithm 1 are very nice to understand the paper).

---

- Significance

This paper is really nice to be adapted to online optimization of any continuous hyperparameters. Further, it is easy to be combined  with other hyperparameter optimization techniques, since the main modification is the meta-objective; for example, one can use implicit differentiation [1] to obtain $\nabla Q(\phi)$ instead of using automatic differentiation of 1-step lookahead.

---

- Weakness

I think this paper is really good, but one thing to improve this paper (I am not going to degrade my score if authors would not address it) is that it is worthwhile to compare it with other hyperparameter optimization methods such as Hypergradient Descent [2]. I think the comparison would provide a clear insight to understand the effectiveness of the proposed meta-objective, since the algorithm is quite similar to [2] except that the discrepancy terms.

[1] Jonathan Lorraine, Paul Vicol, and David Duvenaud. Optimizing millions of hyperparameters by implicit differentiation. In International Conference on Artificial Intelligence and Statistics, 464 pages 1540–1552. PMLR, 2020.

[2] Atilim Gunes Baydin, Robert Cornish, David Martinez Rubio, Mark Schmidt, and Frank Wood. Online learning rate adaptation with hypergradient descent. arXiv preprint arXiv:1703.04782, 2017.

---

> ### Author Response · Authors · 2022-08-02
> **Response to Reviewer NLDo**
>
> **Q: It is worthwhile to compare it with other hyperparameter optimization methods such as Hypergradient Descent [1].**
>
> Thank you for the suggestions! We have added comparisons to Hypergradient Descent (HD) (Li et al., 2017) and L4 (Lee and Choi, 2018)] for the learning rate adaptation task, and First-Order Preconditioning (FOP) (Park and Oliva, 2019) for the preconditioning matrix adaptation task in the updated paper (Appendix D.6, “Comparison to HD, L2, and FOP”). Note that FOP dynamically adapts a meta-learned preconditioning matrix throughout a single training run using the objective motivated by HD.
>
> ### HD and L4 Comparison
>
> Here, we compare APO to HD and L4 as additional baselines that perform online learning rate adaptation. These results are for training ResNet34 on CIFAR-10. Additional results are provided in Appendices D.6.1 and D.6.2.
>
> | Method      | Fixed | Decayed |   HD  |   L4  | APO (Ours) |
> |-------------|:-----:|:-------:|:-----:|:-----:|:----------:|
> | **SGD**     | 93.00 |  93.54  | 92.53 | 85.93 |    94.27   |
> | **SGDm**    | 92.99 |  95.08  | 92.79 | 88.35 |    94.47   |
> | **RMSprop** | 92.87 |  93.87  |   -   | 80.93 |    93.97   |
> | **Adam**    | 93.23 |  94.12  | 92.54 | 85.10 |    93.80   |
>
> ### FOP Comparison
>
> Here, we compare APO-P to FOP as additional baselines that perform preconditioning adaptation. Additional results are provided in Appendix D.6.3.
>
> | Model  |   SGDm   | Adam  | KFAC  | FOP    |   APO-P   |
> |--------|:--------:|:-----:|:-----:|:------:|:---------:|
> | C-10 LeNet    | 75.73   | 73.41 | 76.63 | 75.25 | **77.42** |
> | C-10 AlexNet  | 76.27   | 76.09 | 78.33 | 76.52 | **81.14** |
> | C-10 VGG16    | 91.82   | 90.19 | 92.05 | 91.65 | **92.13** |
> | C-10 ResNet-18 | 93.69  | 93.27 | 94.60 | 93.76 | **94.75** |
> | C-10 ResNet-32 | 94.40  | 93.30 | 94.49 | 93.90 | **94.83** |
> | C-100 AlexNet  | 43.95   | 41.82 | 46.24 | 44.66 | **52.35** |
> | C-100 VGG16    | 65.98   | 60.61 | 61.84 | 61.64 | **67.95** |
> | C-100 ResNet-18 | 76.85  | 70.87 | 76.48 | 75.93 | **76.88** |
> | C-100 ResNet-32 | **77.47**  | 56.58 | 75.70 | 75.66 |  77.41    |
>
>
> **Q: This paper uses an efficient approximation for optimizing the meta-objective, by considering only a 1-step horizon. If one updates the meta-parameter for every $K$-step, it is natural for me to unroll all the previous $K$ steps (if it is possible). For example, either FMD (RMD is possible for preconditioning matrix) or implicit differentiation are feasible to unroll them. Can you elaborate on the reason why one should use a 1-step approximation?**
>
> Theorem 1 shows that exact optimization of the 1-step lookahead meta-objective can recover classic first- and second-order optimization algorithms, motivating our meta-objective. While $K$-step lookahead would certainly be possible, we leave this investigation for future work. A key motivation for limiting the number of lookahead steps $K$ is to maintain good compute and memory costs. Differentiating through a $K$-step unroll (to compute the meta-gradient) requires storing $K$ copies of the neural net parameters in memory, which may be prohibitive for large models. In addition, a $K$-step unroll requires $K$ forward passes (and a more expensive backward pass for the meta-gradient); in order to keep the computational overhead low, one would have to perform meta-updates $K$ times less frequently, likely hurting effectiveness—even with infrequent meta-updates, the memory cost of $K$-step unrolling may be intractable for large models.
>
> Note that implicit differentiation is not applicable to optimizing optimization hyperparameters. Broadly, there are at least two classes of hyperparameters: regularization hyperparameters (such as $L_2$ regularization and data augmentation) and optimization hyperparameters (such as learning rates and preconditioners). A crucial difference between them is that regularizers affect the underlying training loss manifold (e.g., they change the set of fixpoints), while optimizer hyperparameters affect how we traverse across the (fixed) training loss manifold. Implicit differentiation is only applicable to hyperparameters which affect the fixpoints. We compare hyperparameter optimization approaches conceptually in Table 6, Appendix B.
>
> -------
>
> [1] Baydin et al., “Online Learning Rate Adaptation with Hypergradient Descent,” ICLR 2018.
>
> [2] Rolinek and Martius, “L4: Practical Loss-Based Stepsize Adaptation for Deep Learning,” NeurIPS 2018.
>
> [3] Moskovitz et al., “First-Order Preconditioning via Hypergradient Descent,” arXiv 2019.

---

### Official Review · Reviewer_4GgD · 2022-07-13

**Rating:** 7
**Confidence:** 4
**Soundness:** 4 excellent
**Presentation:** 4 excellent
**Contribution:** 3 good

**Summary:**

This paper proposes Amortized Proximal Optimization method, which parameterizes proximal point method and performs online meta-learning the parameters during training. Specifically, they emphasize the importance of considering both function space discrepancy and weight space discrepancy at the same time. They consider two meta-learning scenarios: 1) learning rate scheduling and 2) Kronecker-factored preconditioning matrix. Meta-learning is done with usual one-step lookahead method, but w.r.t. the same training minibatch. The experimental results demonstrate the efficacy of the proposed method over diverse learning scenarios, in terms of training convergence and generalization performance.

**Questions:**

- How the hyperparameters $\lambda_{FSD}$ and $\lambda_{WSD}$ have been selected?
- Why in the outer-loop you used the same minibatch to compute the loss, which seems a little bit unusual?
- Typo: L279, "Figure 4 (right)" --> "Figure 4 (middle)"

**Limitations:**

They did not particularly mentioned the limitations of this work.

**Strengths And Weaknesses:**

# Strengths
- paper is very clearly written and easy to follow
- Nice introduction motivating why we need both FSD and WSD at the same time in Figure 1, which seems to be the main contribution of this paper.
- Connection between the proposed method and the existing second-order optimization methods is well explained.
- Computationally efficient parameterization for the preconditioning matrix, as well as reasonable one-step lookahead meta-learning procedure
- Experiments are very extensive, and overall achieve competitive performance compared to the existing optimizers without learnable parameters.

# Weaknesses
- The biggest and significant weakness of this paper is that they did not compare with any of the existing learnable optimizers. Although the existing meta-learned optimizers mostly consider offline setting (Li et al, Lee et al, Park et al, Flennerhag et al.), Some recent work consider online setting such as Micaelli et al., and Gao et al. Especially, Micaelli et al. proposed to perform online optimization for learning rate, weight decay parameter, and momentum parameter at the same time by using forward mode differentiation, which seems very relevant to this submission. I strongly encourage the authors to compare their model to Micaelli et al. at least for the learning rate scheduling scenario. Gao et al. also seems very relevant to this work as they parameterize Bregman divergence of mirror descent. I believe there should be much more literatures that I'm not aware of, so I will defer it to the discussions with other reviewers. Nonetheless, I believe that the authors should compare with at least one or two existing learnable optimizers. I will adjust my score (either decrease or increase) based on this.


# References
- Li et al, Meta-SGD: Learning to Learn Quickly for Few-Shot Learning, 2017
- Lee et al, Gradient-based Meta-learning with Learned Layerwise Metric and Subspace, ICML 2018
- Park et al, Meta-Curvature, NeurIPS 2019
- Flennerhag et al, Meta-Learning with Warped Gradient Descent, ICLR 2020
- Micaelli et al, Gradient-based Hyperparameter Optimization Over Long Horizons, NeurIPS 2021
- Gao et al., Meta Mirror Descent: Optimiser Learning for Fast Convergence, ICLR 2022 workshop

---

> ### Author Response · Authors · 2022-08-02
> **Response to Reviewer 4GgD (Part 1)**
>
> **Q: Comparisons to existing learnable optimizers.**
>
> Thank you for the references! We have added citations to the ones we missed, in the updated paper. We have added discussions of these related works to Appendix B (Extended Related Work), as well as a table comparing many meta-optimization algorithms to clarify the scope of APO and its positioning in the literature (Table 6 in Appendix B).
>
> For each method, we consider: 1) whether it scales to high-dimensional meta-parameters—most gradient-based methods do, except those that rely on forward-mode auto-diff; 2) whether it dynamically adapts the meta-parameters online or only learns a fixed meta-parameter used for a full training run; 3) whether it requires a task distribution for training, or can operate on a single task of interest; 4) whether it operates on short unrolls of the training procedure (as opposed to requiring complete training runs for each meta-parameter update); and 5) which meta-parameters it can tune.
>
> | Method                | High-Dim | Online | No Task Dist. | Short Unrolls | Meta-Params                        |
> |-----------------------|:--------:|:------:|:-------------:|:-------------:|------------------------------------|
> | **Random Search**     |     ✘    |    ✘   |       ✔       |       ✘       | Any hparam                         |
> | **PBT**               |     ✘    |    ✔   |       ✔       |       ✔       | Any hparam                         |
> | **BayesOpt**          |     ✘    |    ✘   |       ✔       |       ✘       | Any hparam                         |
> | **BPTT**              |     ✔    |    ✔   |       ✔       |       ✘       | Differentiable hparams             |
> | **RTHO**              |     ✘    |    ✔   |       ✔       |       ✔       | Differentiable hparams             |
> | **IFT**               |     ✔    |    ✔   |       ✔       |       ✔       | Diff. regularization hparams       |
> | **L4**                |     ✘    |    ✔   |       ✔       |       ✔       | LR                                 |
> | **FDS**               |     ✘    |    ✘   |       ✔       |       ✘       | LR, momentum, WD                   |
> | **Meta-SGD**          |     ✘    |    ✘   |       ✘       |       ~       | Diagonal preconditioner            |
> | **Meta-Curvature**    |     ✔    |    ✘   |       ✘       |       ~       | Preconditioner via tensor products |
> | **WarpGrad**          |     ✔    |    ✘   |       ✘       |       ✔       | Preconditioning "warp-layers"      |
> | **MetaMD**            |     ✔    |    ~   |       ✘       |       ✘       | Bregman divergence                 |
> | **Learned Optimizer** |     ✔    |    ✔   |       ✘       |       ✘       | Learned optimizer parameters       |
> | **HD**                |     ✘    |    ✔   |       ✔       |       ✔       | LR                                 |
> | **FOP**               |     ✔    |    ✔   |       ✔       |       ✔       | Preconditioner                     |
> | **APO (Ours)**        |     ✔    |    ✔   |       ✔       |       ✔       | Preconditioner, LR                 |
>
> Many approaches that have been used to meta-learn preconditioning matrices require a distribution of tasks for meta-training, including Meta-SGD, Meta-Curvature and MT-nets. In addition, these methods learn a fixed preconditioning matrix that is not adapted online over the course of a training run. In contrast, APO operates within a single training run, and does not require a distribution of tasks; APO also adapts the learned preconditioner online during training.
>
> The most conceptually-similar approach to APO-Precond is First-Order Preconditioning (FOP) (Moskovitz et al., 2019), which also dynamically adapts a meta-learned preconditioning matrix over the course of a single training run. Like APO, FOP directly learns the preconditioner and does not require matrix inversion. However, there are two key differences: (1) FOP uses a different meta-objective following Hypergradient Descent (HD) (Lee and Choi, 2018), that does not include WSD or FSD terms and that evaluates the loss on a separate mini-batch than was used to compute the gradient; and (2) FOP parameterizes the preconditioner as $MM^\top$ for an unconstrained matrix $M$  (separately for each layer), while APO uses a more efficient and structured parameterization inspired by EKFAC.

---

> > ### Author Response · Authors · 2022-08-02
> > **Response to Reviewer 4GgD (Part 2)**
> >
> > **Q: Comparisons to existing learnable optimizers. (Continued)**
> >
> > We would like to note that Micaelli et al. (2021) consider a very different setting than ours; they do not perform online hyperparameter optimization, but rather an offline optimization. Their approach (FDS) performs forward-mode gradient accumulation over an entire training run (or a substantial subset of it), and only makes a single hyperparameter update per run. Due to its forward-mode auto-diff, it is not applicable to high-dimensional hyperparameters such as the preconditioning matrix, which our method can tune. In addition, for learning rate optimization, Micaelli et al. (2021) only tune either a fixed learning rate or the parameters of a manually-specified schedule parameterization, while APO adapts the learning rate dynamically without a pre-specified schedule type. Two methods more related to APO-LR are Hypergradient Descent (HD) (Baydin et al., 2018) and L4 (Rolinek and Martius, 2018).
> >
> > We agree with the reviewer that a comparison to the existing learning rate and preconditioner adaptation methods would strengthen our paper. In the revised version of our manuscript, we have added empirical comparisons to Hypergradient Descent (HD) and L4 for learning rate adaptation and First-Order Preconditioning (FOP) (Moskovitz et al., 2019) for preconditioning adaptation. We have included these comparisons in a new appendix section, Appendix D.6.
> >
> >
> > ### HyperGradient Descent
> >
> > The meta-objective used in Hypergradient Descent (HD) is the expected value of the training loss after a parameter update, $\mathbb{E}_{\mathcal{B} \sim \mathcal{D}}[\mathcal{J}(u(\boldsymbol{\theta}, \boldsymbol{\phi}))]$. This is the same objective that was analyzed theoretically by Wu et al. (2018), that was shown to lead to sub-optimal behavior when the objective is stochastic. In particular, this meta-objective encourages the learning rate to drop rapidly, which reduces the expected loss in the short-term by eliminating fluctuations in the loss caused by random mini-batch sampling, at the cost of worse long-term behavior. We performed experiments with HD, and found that it suffers from the short horizon bias issue; the only way to prevent the LR from reducing too quickly is to use a very small meta-learning rate, which becomes more heuristic. Empirically, the APO meta-objective does not suffer from this issue, allowing us to use larger meta-learning rates without having the LR decrease too quickly. Also, because the meta-learning rate for HD must be small, HD is not robust to the initial learning rate; the initial LR must be tuned in order to obtain decent performance. In contrast, APO is very robust to the initial learning rate, behaving nearly identically for initial learning rates spanning 6 orders of magnitude, from 1e-1 to 1e-7 (shown in Appendix J.2).
> >
> > We have added a discussion and empirical comparison with HD in Appendix D.6.1, along with figures that show how using a larger meta-learning rate in HD leads to small adapted LRs.
> >
> > | Method      | Fixed | Decayed |   HD  |   L4  | APO (Ours) |
> > |-------------|:-----:|:-------:|:-----:|:-----:|:----------:|
> > | **SGD**     | 93.00 |  93.54  | 92.53 | 85.93 |    94.27   |
> > | **SGDm**    | 92.99 |  95.08  | 92.79 | 88.35 |    94.47   |
> > | **RMSprop** | 92.87 |  93.87  |   -   | 80.93 |    93.97   |
> > | **Adam**    | 93.23 |  94.12  | 92.54 | 85.10 |    93.80   |
> >
> > ### First-Order Preconditioning
> >
> > The First-Order Preconditioning (FOP) adapts a meta-learned preconditioning matrix throughout a single training run with the meta-objective similar to that of HD. Due to having similar meta-objectives, FOP also suffers from the same problem as HD mentioned above. Moreover, FOP uses $MM^\top$ style preconditioner parameterization, which has significantly higher memory overhead compared to APO-P. We empirically evaluated APO-P and FOP on image reconstruction and classification tasks. APO-P achieved better convergence in reconstruction experiments and better generalization performance in the image classification experiments.
> >
> > We have added an empirical comparison with FOP in Appendix D.6.3.
> >
> > | Model  |   SGDm   | Adam  | KFAC  | FOP    |   APO-P   |
> > |--------|:--------:|:-----:|:-----:|:------:|:---------:|
> > | C-10 LeNet    | 75.73   | 73.41 | 76.63 | 75.25 | **77.42** |
> > | C-10 AlexNet  | 76.27   | 76.09 | 78.33 | 76.52 | **81.14** |
> > | C-10 VGG16    | 91.82   | 90.19 | 92.05 | 91.65 | **92.13** |
> > | C-10 ResNet-18 | 93.69  | 93.27 | 94.60 | 93.76 | **94.75** |
> > | C-10 ResNet-32 | 94.40  | 93.30 | 94.49 | 93.90 | **94.83** |
> > | C-100 AlexNet  | 43.95   | 41.82 | 46.24 | 44.66 | **52.35** |
> > | C-100 VGG16    | 65.98   | 60.61 | 61.84 | 61.64 | **67.95** |
> > | C-100 ResNet-18 | 76.85  | 70.87 | 76.48 | 75.93 | **76.88** |
> > | C-100 ResNet-32 | **77.47**  | 56.58 | 75.70 | 75.66 |  77.41    |

---

> > > ### Author Response · Authors · 2022-08-02
> > > **Response to Reviewer 4GgD (Part 3)**
> > >
> > > **Q: How the hyperparameters $\lambda_{\text{FSD}}$ and $\lambda_{\text{WSD}}$ have been selected?**
> > >
> > > We performed a grid search over the weight-space and function-space lambda coefficients, which we described in Appendix C ("Experimental Details") for each experiment. The search space size for $\lambda_{\text{FSD}}$ and $\lambda_{\text{WSD}}$ was generally similar to learning rate search space size for first-order baseline optimizers. For example, in the image classification task (ResNet-18 & CIFAR-10), APO-P searched over 8 different $\lambda$ configurations, and SGDm searched over 8 different learning rate configurations. We also include ablation studies over these hyperparameters in Appendix J (“Ablations”) and show that APO is not highly sensitive to $\lambda_{\text{FSD}}$ and $\lambda_{\text{WSD}}$.
> > >
> > >
> > > **Q: Why in the outer-loop you used the same minibatch to compute the loss, which seems a little bit unusual?**
> > >
> > > Using a random minibatch to evaluate the loss after taking a lookahead step corresponds to the meta-objective analyzed in Wu et al., 2018 (which we discuss in Appendix B (“Extended Related Work”)). This meta-objective was shown to yield undesirable behavior, where the learning rate decreases rapidly to reduce fluctuations of the loss caused by stochastic mini-batch evaluation. Evaluating the lookahead step on the same mini-batch that was used to compute the gradient for the step avoids this issue, and leads to larger learning rates; the function-space and weight-space discrepancy terms regularize the step such that it does not move too far, e.g., decreasing the loss on the current mini-batch but altering predictions on other mini-batches. We discuss this algorithmic choice in Appendix E (“Importance of Using the Same Mini-batch”).
> > >
> > > Moreover, using the same mini-batch is essential for the proof of Theorem 1; the use of the same mini-batch and assuming the linearized loss term and quadratic FSD term allow our meta-objective to find a preconditioner that recovers many existing second-order optimization methods, $\mathbf{P}^{\star} = (\lambda_{\text{FSD}} \mathbf{G} + \lambda_{\text{WSD}} \mathbf{I})^{-1}$.
> > >
> > >
> > > **Q: Typo: L279, Figure 4 (right) --> Figure 4 (middle)**
> > >
> > > Thank you for your careful reading! We fixed the typo in the updated manuscript.
> > >
> > > ---------------
> > >
> > > [1] Li et al., “Meta-SGD: Learning to Learn Quickly for Few-Shot Learning,” arXiv 2017.
> > >
> > > [2] Lee and Choi, “Gradient-based Meta-learning with Learned Layerwise Metric and Subspace,” ICML 2018.
> > >
> > > [3] Park and Oliva, “Meta-Curvature,” NeurIPS 2019.
> > >
> > > [4] Flennerhag et al., “Meta-Learning with Warped Gradient Descent,” ICLR 2020.
> > >
> > > [5] Micaelli et al., “Gradient-based Hyperparameter Optimization Over Long Horizons,” NeurIPS 2021.
> > >
> > > [6] Gao et al., “Meta Mirror Descent: Optimiser Learning for Fast Convergence,” ICLR 2022 Workshop.
> > >
> > > [7] Baydin et al., “Online Learning Rate Adaptation with Hypergradient Descent,” ICLR 2018.
> > >
> > > [8] Moskovitz et al., “First-Order Preconditioning via Hypergradient Descent,” arXiv 2019.
> > >
> > > [9] Wu et al., “Understanding Short-Horizon Bias in Stochastic Meta-Optimization,” ICLR 2018.
> > >
> > > [10] Rolinek and Martius, “L4: Practical Loss-Based Stepsize Adaptation for Deep Learning,” NeurIPS 2018.

---

> > > > ### Comment · Reviewer_4GgD · 2022-08-02
> > > > **Thanks for the detailed comments**
> > > >
> > > > I appreciate the author's efforts for adding more baselines and clarifications. I thought Micaelli et al. performs online learning, but maybe I misunderstood. I will read it again carefully. The responses are generally satisfactory especially that the authors add more baselines that performs online hyperparameter optimization. Overall the paper looks good and experimental results are very extensive.
> > > >
> > > > Another limitation of this paper is that the range of hyperparameters are only confined within LR and preconditioner. If authors could show that the method can be applied more generally (at least one more scenario), then I would be happy to further increase my score.
> > > >
> > > > Thank you!

---

> > > > > ### Author Response · Authors · 2022-08-09
> > > > > **Tuning other optimization parameters**
> > > > >
> > > > > Thank you very much for your helpful comments! We have added proof-of-concept experiments in Appendix Section D.7, where we use APO to tune two hyperparameters of RMSprop that are rarely tuned by hand: 1) the value of $\epsilon$ used in the denominator of the RMSprop update; and 2) the power to which the running average of the squared gradients is raised, which we denote $\rho$. We describe the RMSprop update rule in detail in Appendix D.7.
> > > > >
> > > > > We investigated the effect of tuning $\\{ \epsilon, \rho \\}$ using APO on two tasks: 1) training a two-hidden-layer MLP with 1000 hidden units per layer on FashionMNIST; and 2) training ResNet32 on CIFAR-10. We set the learning rate to the best fixed value for each task: $\eta = 0.001$ for CIFAR-10 and $\eta=0.0001$ for FashionMNIST. We compared to the RMSprop baseline with the same optimal fixed learning rates ($\eta=0.001$ for CIFAR-10 and $\eta=0.0001$ for FashionMNIST) and with the default values $\epsilon = \text{1e-8}$ and $\rho = 0.5$. The results for FashionMNIST and CIFAR-10 are shown in Appendix D.7 Figures 32 and 33, respectively. We found that on both of these tasks, tuning $\\{\epsilon, \rho \\}$ resulted in faster training and lower final training loss.

---

> > > > > > ### Comment · Reviewer_4GgD · 2022-08-10
> > > > > > **Thanks for the additional experiments**
> > > > > >
> > > > > > I appreciate the authors' efforts for those additional experiments. The results look satisfactory. Actually my original intention was to check whether APO can be applied to other types of hyperparameters than optimizer parameters, but the RMSProp experiments seem interesting as well. I'm keeping my score, but can be adjusted favorably after the upcoming discussion phase.
> > > > > >
> > > > > > One additional concern is wall-clock convergence speed. Although one-step lookahead differentiation is efficient, it would be still slower in wall-clock time than conventional non-learnable optimizers. But I think it is less of a concern because APO achieves lower training and test loss in the end, regardless its convergence speed. Showing this may not be necessary, but will improve the paper a little bit.
> > > > > >
> > > > > > (edit: I just found the wall-clock results in Figure 10. So just forget about it!)
> > > > > >
> > > > > > Thanks, and have a nice day!

---

### Author Response · Authors · 2022-08-02
**Updated the Paper, Highlighted Additions in Blue**

We thank all the reviewers for their thoughtful reviews and helpful comments. We appreciate that the reviewers found the paper to be well-written (4GgD, NLDo, nHHL) and well-motivated with connections between proximal optimization and first- and second-order optimizers (4GgD, NLDo, nHHL), the meta-objective to be novel and the algorithm to be efficient (4GgD, NLDo), and the experiments to be extensive (4GgD, NLDo, nHHL). We respond to each reviewer below, and summarize the updates we have made to the paper here.


## Paper Updates

We have revised the paper PDF to address the reviewers’ questions, with the updates highlighted in ${\color{blue} \text{blue text}}$. We have added:

* Experiments comparing to Hypergradient Descent (HD) [1] for online learning rate adaptation in the new Appendix D.6.1.
* Experiments comparing to L4 [2], another approach for online LR adaptation, in the new Appendix D.6.2.
* Experiments comparing to First-Order Preconditioning (FOP) [3], a method for tuning the preconditioning matrix online, in the new Appendix D.6.3.
* A discussion of more related work in Appendix B, as well as the new Table 6 summarizing the conceptual differences between many hyperparameter optimization and meta-learning methods.
* Extended tables in Appendix D with standard deviations reported for each approach.


### Comparison to Hypergradient Descent and L4

We compared APO to the HD and L4 baselines for ResNet32 and ResNet34 on CIFAR-10 (the table below shows results for ResNet34, more results are in Appendix D.6.1 and D.6.2).

| Method      | Fixed | Decayed |   HD  |   L4  | APO (Ours) |
|-------------|:-----:|:-------:|:-----:|:-----:|:----------:|
| **SGD**     | 93.00 |  93.54  | 92.53 | 85.93 |    94.27   |
| **SGDm**    | 92.99 |  95.08  | 92.79 | 88.35 |    94.47   |
| **RMSprop** | 92.87 |  93.87  |   -   | 80.93 |    93.97   |
| **Adam**    | 93.23 |  94.12  | 92.54 | 85.10 |    93.80   |

### Comparison to First-Order Preconditioning
We compared APO-Precond to the FOP baseline for a variety of network architectures on CIFAR-10 and CIFAR-100 (Appendix D.6.3).

| Model  |   SGDm   | Adam  | KFAC  | FOP    |   APO-P   |
|--------|:--------:|:-----:|:-----:|:------:|:---------:|
| C-10 LeNet    | 75.73   | 73.41 | 76.63 | 75.25 | **77.42** |
| C-10 AlexNet  | 76.27   | 76.09 | 78.33 | 76.52 | **81.14** |
| C-10 VGG16    | 91.82   | 90.19 | 92.05 | 91.65 | **92.13** |
| C-10 ResNet-18 | 93.69  | 93.27 | 94.60 | 93.76 | **94.75** |
| C-10 ResNet-32 | 94.40  | 93.30 | 94.49 | 93.90 | **94.83** |
| C-100 AlexNet  | 43.95   | 41.82 | 46.24 | 44.66 | **52.35** |
| C-100 VGG16    | 65.98   | 60.61 | 61.84 | 61.64 | **67.95** |
| C-100 ResNet-18 | 76.85  | 70.87 | 76.48 | 75.93 | **76.88** |
| C-100 ResNet-32 | **77.47**  | 56.58 | 75.70 | 75.66 |  77.41    |

We thank the reviewers again for their time!

[1] Baydin et al., “Online Learning Rate Adaptation with Hypergradient Descent,” ICLR 2018.

[2] Rolinek and Martius, “L4: Practical Loss-Based Stepsize Adaptation for Deep Learning,” NeurIPS 2018.

[3] Moskovitz et al., “First-Order Preconditioning via Hypergradient Descent,” arXiv 2019.

---

### Meta-Review · Area_Chair_sjWU · 2022-08-28

**Recommendation:** Accept
**Confidence:** Certain

**Metareview:**

This paper proposes a technique for online adaptation of optimization hyper-parameters. The key advance seems to be the simultaneous minimization of function and weight space discrepancy, in addition to the minimization of the minibatch loss.

All reviewers recommended this paper be accepted. One reviewer raised their score by two points, and went on to "moderately champion" the paper during discussion. The paper appears to have additionally meaningfully improved through the course of the review process.

Based upon the reviewer consensus, I also recommend paper acceptance.

**Award:**

No

---

### Decision · Program_Chairs · 2022-09-14

Accept